# Structure determination from single molecule X-ray scattering with three photons per image

Benjamin von Ardenne[1], Martin Mechelke[1] & Helmut Grubmüller[1]

Scattering experiments with femtosecond high-intensity free-electron laser pulses provide a new route to macromolecular structure determination. While currently limited to nano-crystals or virus particles, the ultimate goal is scattering on single biomolecules. The main challenges in these experiments are the extremely low signal-to-noise ratio due to the very low expected photon count per scattering image, often well below 100, as well as the random orientation of the molecule in each shot. Here we present a de novo correlation-based approach and show that three coherently scattered photons per image suffice for structure determination. Using synthetic scattering data of a small protein, we demonstrate near-atomic resolution of 3.3 Å using $3.3 \times 10^{10}$ coherently scattered photons from $3.3 \times 10^{9}$ images, which is within experimental reach. Further, our three-photon correlation approach is robust to additional noise from incoherent scattering; the number of disordered solvent molecules attached to the macromolecular surface should be kept small.

---

[1] Department of Theoretical and Computational Biophysics, Max Planck Institute for Biophysical Chemistry, Am Fassberg 11, 37077 Göttingen, Germany. Correspondence and requests for materials should be addressed to H.Gül. (email: hgrubmu@mpibpc.mpg.de)

First proposed by Neutze et al.[1], single-particle scattering experiments with high-intensity X-ray free-electron lasers (XFELs) hold the promise to solve the three-dimensional atomic structure of biological macromolecules such as proteins without the need for crystallization[2–5]. High-repetition femtosecond X-ray pulses are used to outrun the severe radiation damage due to Auger decay and Coulomb explosion and thus allow for extremely high peak brilliance pulses to the point where single molecules can be imaged. Indeed, the first proof of principle experiments[6,7] determined the 3D structure of single mimivirus particles to a resolution of 125 nm and Hosseinizadeh et al. recently demonstrated the structure determination of a coliphage virus with 9 nm resolution[8]. In these experiments, more than $10^7$ photons per X-ray pulse were scattered by the virus and recorded on a pixel detector (Fig. 1a). In contrast, for a medium-sized molecule and an expected XFEL fluence of $1.3 \times 10^6$ photons $nm^{-2}$ ($10^{12}$ photons) at a 1 µm focus diameter[9], only about 10–50 coherently scattered photons per scattering image are expected at a beam energy of 5 keV (2.5 Å wavelength)[9–11].

The high statistical noise in this extreme Poisson regime poses considerable methodological challenges, and hence XFEL structure determination attempts almost exclusively focus on nanocrystals[12–18]. A particular challenge is to determine the orientation of the molecule for each image to assemble all recorded images in 3D Fourier space for subsequent electron density determination. For macroscopic 2D objects and 3D objects rotated around a single axis, Philipp et al. showed structure recovery from only 2.5 photons per image on average[19–21], but the method was not extended or applied to three-dimensional objects or molecules with unknown orientation. For single-molecule scattering experiments, several orientation determination methods were developed[22–29], which, however, require at least 100 photons per image. Alternatively, manifold reconstruction algorithms (manifold embedding)[30–33] forego the explicit assembly in Fourier space and instead use the similarity between scattering images to determine the manifold of orientations. However, also for these methods, successful structure determination was only reported for much more than 100 photons per image.

In fluorescence microscopy or cryo-electron microscopy, time integrated and time-correlated single-photon counting is used at extremely low signal-to-noise ratios[34]. In the context of single-molecule X-ray scattering, two-photon correlations were successfully used to determine the molecular shape of symmetric particles[35,36] and the structure of particles randomly oriented around one axis[37,38]. However, two photons are not sufficient to retrieve the structure de novo.

Based on early analytic work on degenerate three-photon correlations[39], structure determination of mesoscopic cylindrical particles[40] and of a highly symmetric icosahedral virus[41,42] was demonstrated. This approach is limited to only a small fraction of the recorded correlations; however, also this method has so far not been applied to de novo single-molecule structure determination.

Here, we use the full three-photon correlation as an orientation-independent representation of the scattering images. We demonstrate that only three coherently scattered photons per image are required for de novo structure determination, such that near-atomic resolution for single biomolecules should in principle be possible even at extremely low photon counts.

## Results

**Structure determination**. Like in X-ray crystallography, the photon distribution of each scattering image follows the intersection between the Ewald sphere and the 3D intensity, $I(\mathbf{k}) \propto |\mathcal{FT}[\rho(\mathbf{x})]|^2$, which is proportional to the absolute square of the Fourier-transformed electron density $\rho(\mathbf{x})$. The orientation of the Ewald sphere depends on the molecular orientation and so does the scattering image. In contrast to X-ray crystallography, $I(\mathbf{k})$ is continuous for single-molecule scattering, rendering the phase problem accessible to established methods[43–46]. Because the orientation of the molecule is unknown, here $I(\mathbf{k})$ is determined via the three-photon correlation function $t(k_1, k_2, k_3, \alpha, \beta)$ which is accumulated from all photon triplets in the recorded scattering images as illustrated in Fig. 1b.

To recover $I(\mathbf{k})$, an analytic expression of the full three-photon correlation as a function of the 3D intensity $I(\mathbf{k})$ was derived using shell-wise spherical harmonics (SH) expansions[47] for $I(\mathbf{k}) = \sum_{lm} A_{lm}(|\mathbf{k}|) Y_{lm}(\theta, \varphi)$ (Methods and Supplementary Notes 1–3). This choice allows for adapting the number $K(L^2 + 3L + 2)/2$ of SH basis functions to the target resolution via the largest considered wave number $k_{cut}$, the number $K$ of used shells between 0 … $k_{cut}$, and the expansion order $L$. We were unable to invert the analytic expression of the three-photon correlation, and the number of unknowns (e.g., 4940 for $K = 26$, $L = 18$) is too large for a straightforward numeric solution. To circumvent this problem, we used a probabilistic approach and solved for those SH coefficients $\{A_{lm}(k)\}$ that maximize the probability, $p(\{k_1^i, k_2^i, k_3^i, \alpha^i, \beta^i\} | \{A_{lm}(k)\}) = \prod_{i=1...T} \tilde{t}(k_1^i, k_2^i, k_3^i, \alpha^i, \beta^i)_{\{A_{lm}(k)\}}$ (Bayesian with uniform prior), of observing all $T$ recorded triplets (Methods and Supplementary Notes 4 and 5). Due to their statistical independence, $p$ is the product of the probabilities of

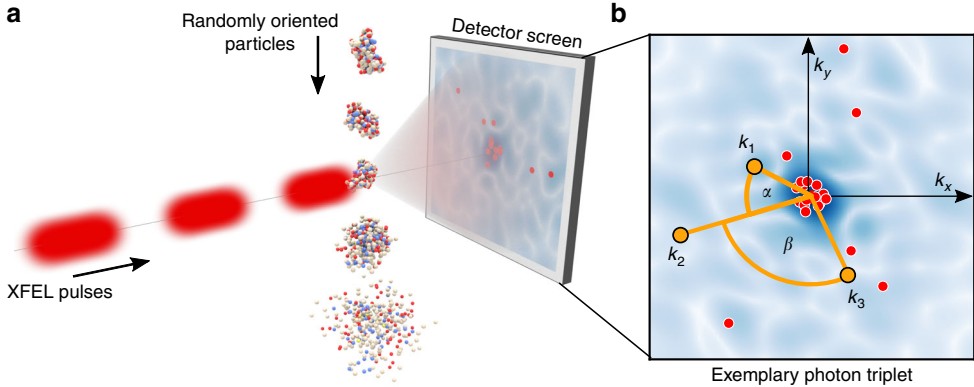

**Fig. 1** Single-molecule scattering and photon correlations. **a** A stream of randomly oriented particles is injected into the XFEL beam, hit sequentially by femtosecond X-ray pulses, and the coherently scattered photons (red dots) are recorded on the pixel detector. **b** In the detector plane $k_x k_y$ the recorded photons are grouped into triplets, each of which is characterized by distances $k_1, k_2, k_3$ to the detector center (orange lines) and the angles $\alpha$ and $\beta$ between the respective photons (orange circular arcs)

observing each recorded photon triplet which is given by the normalized three-photon correlation $\tilde{t}(k_1, k_2, k_3, \alpha, \beta)_{\{A_{lm}(k)\}}$. The search space was further reduced by utilizing the analytic inversion of the two-photon correlation[39] (Methods and Supplementary Note 6), rendering the problem accessible to Monte Carlo simulated annealing[48]. We found that independent Monte Carlo runs converged to similar intensities (Pearson correlation of 0.99), suggesting that the solution of the inversion of the three photon correlations is unique.

Contrary to intuition, smaller molecules are more demanding than larger ones[24]. We therefore challenged our approach by using the 46 residue comprising Crambin protein, which is known to 0.8 Å resolution[49] (Fig. 2e). We estimated an average of 14 coherently scattered photons per Crambin shot, a number which is achieved, e.g., at the XFEL at DESY using an X-ray intensity of $10^{12}$ photons per pulse at 5 keV and a 1 μm beam diameter. The estimates were calculated with the Condor package by Hantke et al.[10] using a flat-top beam profile. An independent calculation using the SimEx simulation framework for imaging single particles at the European XFEL by Fortmann-Grote et al.[9,11] using a realistic beam profiles yielded similar numbers.

As a conservative test case, and to challenge our method, we generated up to $3.3 \times 10^9$ synthetic scattering images with only 10 photons on average, totaling up to $3.3 \times 10^{10}$ recorded photons (Methods). With an expected XFEL repetition rate of up to 27 kHz[50], and assuming a hit rate of 10%, we expect this data to be collected within a few days (Fig. 3d). As discussed in Supplementary Note 8, the data acquisition time substantially decreases to, e.g., approx. 30 min when on average 100 photons per image are recorded (e.g., by shrinking the beam diameter by a factor of 3 to approx. 300 nm), reducing the total number of required photons by a factor 100 to $3.3 \times 10^8$ (and reducing the number of images by a factor 1000 to $3.3 \times 10^6$). Even for a lower hit rate such as 1%, 300 min would suffice in this case.

From the synthetic scattering images, we performed 20 independent structure determination runs (Methods and Supplementary Fig. 7). For all runs we used an expansion order $L = 18$, $K = 26$ shells and a cutoff $k_{cut} = 2.15 \, \text{Å}^{-1}$ (Supplementary Note 7 discusses the optimal parameters), thus setting the maximum achievable resolution to 2.9 Å. Fig. 2a-c compares the average intensity obtained from these 20 runs (green) with the reference intensity derived from the known X-ray structure (blue). Overall, the shape of the intensity is recovered very well and only minor deviations in the outer shells, where fewer photons are recorded, are present.

To assess the achievable resolution of the determined Fourier intensities, we calculated 20 real space electron density maps using an iterative phase retrieval algorithm[45]. Figure 2d and e compares the average of the 20 retrieved densities (d, green shaded structure) with the the reference electron density (e, blue shaded structure) which has been calculated from the Fourier density (including phases) with same cutoff $k_{cut}$ as (d). The cross-correlation between the two densities is 0.9. The Fourier shell correlation (FSC) between the known reference electron density of Crambin and the retrieved averaged electron density was calculated as a function of the wave number $k$[51] (note that we use $k_{in} = 2\pi/\lambda$ for all wave number calculations). Similar to single-particle electron microscopy[51], the wave number $k_{res}$ at which $\text{FSC}(k_{res}) = 0.5$ was used to estimate the achieved radial resolution $\Delta r = 2\pi/k_{res}$. Here, a near-atomic resolution of 3.3 Å was achieved.

**Resolution as function of number of recorded images.** Next we explored how the achieved resolution depends on the number of observed photons (and triplets, respectively), and hence the

number of recorded images. To this end, electron densities were calculated and averaged as above from $1.3 \times 10^7$ up to $3.3 \times 10^{10}$ photons gathered from images with 10 photons on average ($4.7 \times 10^8$ up to $1.2 \times 10^{12}$ triplets). Figure 3a depicts the respective FSC curves for different photon counts along with the 0.5 cutoff (vertical dashed line) and the corresponding resolutions (inset).

As mentioned before, for $3.3 \times 10^{10}$ photons a near-atomic resolution of 3.3 Å was achieved. Decreasing the number of photons by a factor of 10 decreased the resolution only slightly by 0.4 Å to 3.7 Å, which indicates that very likely fewer than $3.3 \times 10^{10}$ photons suffice to achieve near-atomic resolution. If much fewer photons are recorded, e.g. $1.3 \times 10^7$ ($4.4 \times 10^8$ triplets), the resolution decreased markedly to 14 Å. To address the question how much further the resolution can be increased, we mimicked an experiment with infinite number of photons by determining the intensity from the analytically calculated three-photon correlation using Eq. (3) from the Methods section. As can be seen in Fig. 3a (purple line), the resolution only slightly improved by 0.1 Å to about 3.2 Å indicating that at this point either the expansion order $L$ or insufficient convergence of the Monte Carlo-based structure search became resolution limiting. To distinguish between these two possible causes, we phased the electron density directly from the reference intensity, using the same expansion order $L = 18$ as in the other experiments. The reference intensity is free from convergence issues of the Monte Carlo structure determination and the resulting electron density only includes the phasing errors introduced by the limited angular resolution of the SH expansion in Fourier space. The FSC curve of the optimal phasing (gray dashed) shows only a minor increase in resolution to 3.1 Å indicating that the Monte Carlo search decreases the resolution by 0.1 Å. The remaining 0.2 Å difference to the optimal resolution of 2.9 Å at the given $k_{cut}$ (not shown) is attributed to the finite expansion order $L$ and the corresponding phasing errors. We have also independently assessed the overall phasing error by calculating the intensity shell correlation (ISC) between the intensities of the phased electron densities $I_{phased} = \left| \mathcal{FT}[\rho_{retrieved}] \right|^2$ and the intensities before phasing $I_{retrieved}$ (Methods and Supplementary Fig. 8). As discussed in the Methods section, the phasing method does not markedly deteriorate our structures.

Because a large expansion order $L$ requires a larger number of shells $K$, and, therefore, much larger numbers of unknowns (Supplementary Note 7), the question remains at which point overfitting occurs. To quantify this effect for our sets of images, we calculated the achieved resolution as a function of expansion order $L$ for four different total photon counts $5.1 \times 10^7$, $2.0 \times 10^8$, $8.2 \times 10^8$, and $3.3 \times 10^{10}$ ($1.8 \times 10^9$, $7.1 \times 10^9$, $2.8 \times 10^{10}$, and $1.2 \times 10^{12}$ triplets, respectively) at a fixed number of shells $K = 26$. Indeed, as shown in Fig. 3b, for up to $2.0 \times 10^8$ photons, the obtained three-photon correlation is too noisy to yield an improved resolution when increasing the model detail and for larger $L$, the probability $p$ of the intensity model still increases whereas the resolution decreases again, indicating overfitting. In contrast, for larger photon counts ($>8.2 \times 10^8$), the resolution improves even up to the expansion order $L = 18$ and no overfitting is expected here. However, due to the large parameter space, convergence of the simulated annealing becomes computationally demanding (Supplementary Notes 4, 5, and 7).

**Robustness to noise.** We finally assessed how robust our approach is in the presence of additional experimental noise due to, e.g., incoherent scattering, background radiation, detector noise, or scattering at the unstructured fraction of water molecules that may adhere to the surface of the macromolecules[1]. Since only very few single-molecule scattering experiments have

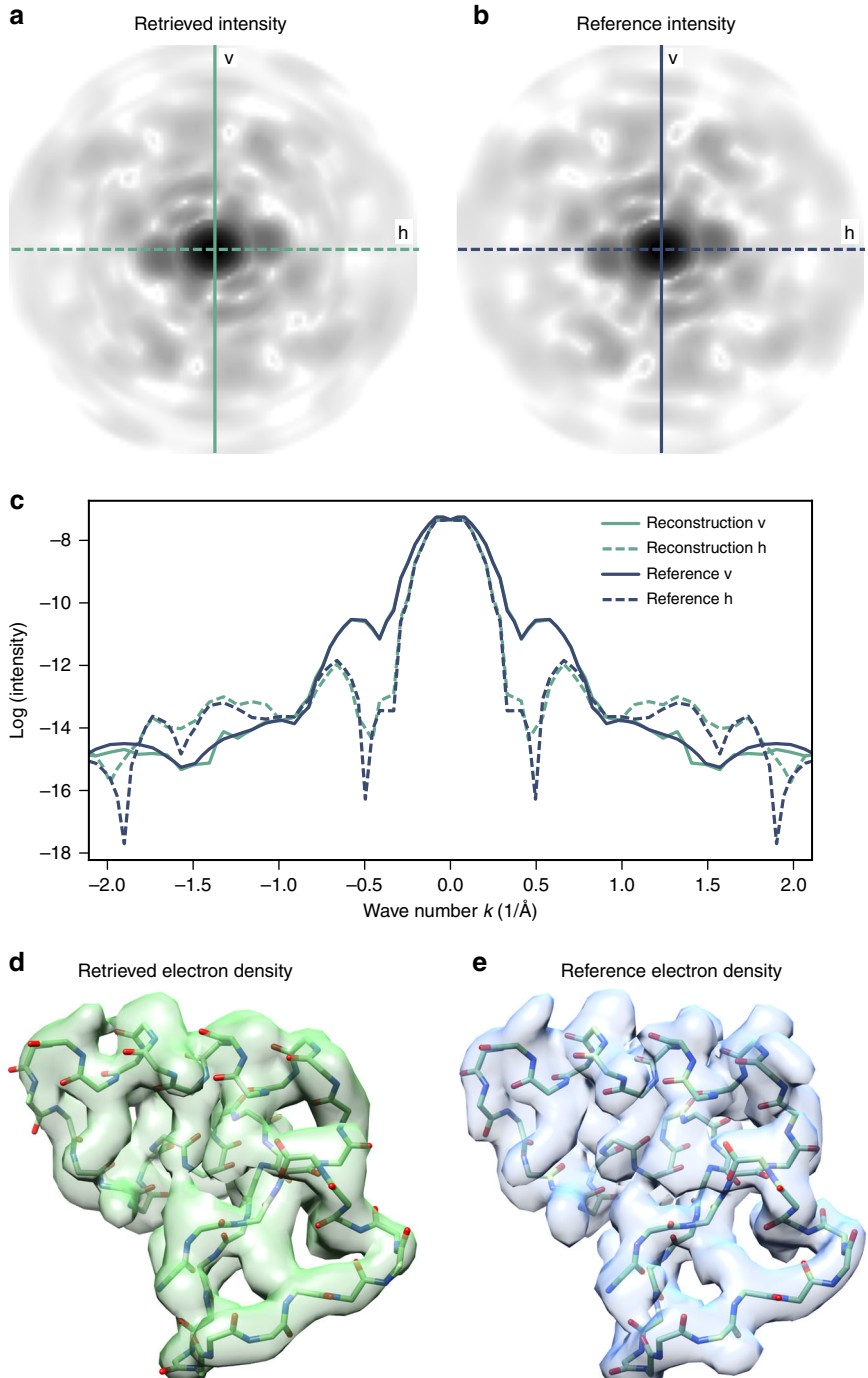

**Fig. 2** Structure determination at 3.3 Å resolution. Comparison of the retrieved density (green lines and structure) and the reference density of Crambin (blue lines and structure) in Fourier space and real space. Shown are averages over 20 structure determination runs, each using the same $3.3 \times 10^9$ images with an average of 10 photons per image yielding $3.3 \times 10^{10}$ photons. A cutoff in reciprocal space of $k_{cut} = 2.15$ Å$^{-1}$ was used and the intensity was expanded with $K = 26$ shells using an expansion order of $L = 18$. **a**, **b** Comparison of the the retrieved intensity (**a**) and the reference intensity (**b**) in the $k_x k_y$-plane (logarithmic shading). **c** Comparison of two orthogonal linear cuts (vertical, v, and horizontal, h) through the $k_x k_y$-planes shown in **a** and **b**. **d**, **e** Comparison of the retrieved electron density (**d**) and the reference electron density (**e**). The latter was calculated from the known Fourier density using the same cutoff $k_{cut} = 2.15$ Å$^{-1}$ in reciprocal space as in **d**. The resolution of the retrieved density is 3.3 Å, the resolution of the reference density is 2.9 Å, and the cross-correlation between the two densities is 0.9

been carried out so far, quantitative noise models are available only for incoherent scattering, for which a noise level of ca. $\gamma = 25\%$[52] is expected. Here we modeled the noise as a Gaussian distribution, $G(k, \sigma) = \gamma (2\pi\sigma^2)^{-1/2} \exp(-k^2/2\sigma^2)$. Depending on the width $\sigma$, different signal-to-noise ratios are expected in the

low-resolution and high-resolution regions of the image, respectively. For incoherent scattering (indicated as gray background) a width of $\sigma = 2.5$ Å$^{-1}$ was assumed[53] (Supplementary Note 9), which corresponds to a relatively uniform noise distribution. Figure 3c (black line) shows a moderate decrease in resolution to

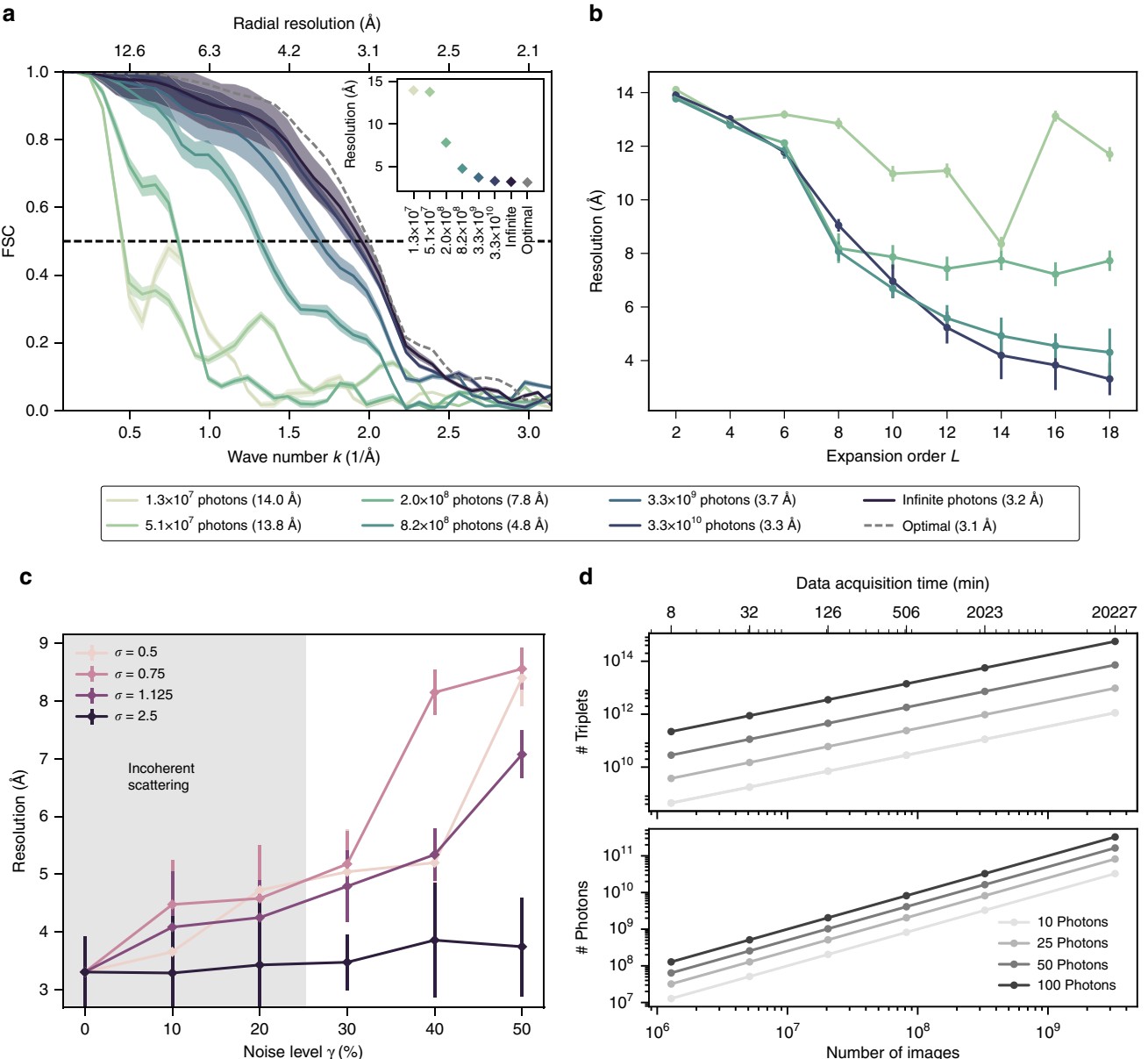

**Fig. 3** Resolution as a function of collected photons and noise. **a** Average Fourier shell correlations (FSC) between the reference electron density of Crambin and the densities retrieved from $1.3 \times 10^7$ to $3.3 \times 10^{10}$ photons ($4.7 \times 10^8$ to $1.2 \times 10^{12}$ triplets) and infinite photon number. As a reference, the optimal FSC is shown (dashed gray), which was calculated directly from the known intensity using the same expansion parameters. The inset shows the corresponding resolutions estimated from $FSC(k_{res}) = 0.5$. The opaque error band was calculated from the standard deviation of the 20 individual FSCs before averaging. **b** Achieved average resolution as a function of the expansion order $L$ using $5.1 \times 10^7$, $2.0 \times 10^8$, $8.2 \times 10^8$, and $3.3 \times 10^{10}$ photons ($1.8 \times 10^9$, $7.1 \times 10^9$, $2.8 \times 10^{10}$, and $1.2 \times 10^{12}$ triplets, respectively). The error bars show the standard error of mean of 20 independent runs. **c** Achieved average resolution for synthetic experiments with $3.3 \times 10^9$ images including an additional fraction $\gamma$ of random photons ($\gamma = 0$–50%) following a Gaussian distribution with varying width $\sigma$. For noise from incoherent scattering (indicated as gray background), we assumed a width $\sigma = 2.5$ Å$^{-1}$, but also included narrower widths $\sigma = [0.5, 0.75, 1.125]$ Å$^{-1}$ as a model for other sources of noise. The error bars show the standard error of mean of 20 independent runs. **d** Expected number of photons and triplets as a function of image numbers for 10, 20, 50, and 100 photons per image. Also shown (top axis) is the estimated data acquisition time given a 27 kHz XFEL repetition rate and 10% hit rate

approx. 3.5 Å when this noise is included within our synthetic experiments (as described in Supplementary Note 9). Additional noise with a uniform distribution from, e.g. background radiation or detector noise, slightly decreased the resolution to 3.8 Å at 50% noise level.

For scattering from disordered water molecules that are attached to the macromolecular surface, a narrower intensity distribution is expected (Supplementary Fig. 4). To also

investigate this effect and the effect of other potential noise sources with non-uniform distribution, in Fig. 3c, we considered noise with widths of $\sigma = [0.5, 0.75, 1.125]$ Å$^{-1}$ and noise levels $\gamma$ between 10 and 50%, the latter corresponding e.g. to up to 100 disordered water molecules per Crambin molecule. The resolution remained better than 5 Å within the 25% noise level but decreases markedly to 9 Å with $\gamma = 50$%, in particular for narrow noise widths of $\sigma = [0.5, 0.75]$ Å$^{-1}$.

**Sample application to experimental data**. To test if our method is also robust against noise in real experimental data, we have determined the structure of the coliphage PR772 virus from the Reddy et al. data set[54] (Supplementary Note 10), albeit at much higher photon counts than our method is targeted for. As described in ref. [54], this image-set has been obtained by filtering the raw images for single molecule hits with diffusion map embedding. Therefore, to mimic low photon counts, we down-sampled the images, which contain over 400,000 photons per image, and generated $3 \times 10^{12}$ triplets using the same rejection sampling method that we used to generate the Crambin images, and subsequently applied the same reconstruction procedure (Supplementary Fig. 5). A resolution of 11.7 nm was achieved, as evidenced from the FSC between two independently determined structures (Supplementary Fig. 6). This resolution is slightly lower than the 9 nm obtained by Hosseinizadeh et al.[55], which may be due to the fact that we used fewer photons, implying additional Poisson noise. Also, in contrast to Hosseinizadeh et al., we have not implicitly imposed any icosahedral symmetry in our reconstructions.

## Discussion

The presented method demonstrates de novo structure determination from as few as three photons per XFEL scattering image at near-atomic resolution. Our synthetic scattering experiments with subsequent structure determination have shown that, for the most challenging case of small biomolecules, a resolution better than 3.3 Å should be achievable with available technology at realistic beam times; specifically, as our conservative estimate rests on a beam fluence of $5.0 \times 10^{11}$ photons per pulse. Assuming a 10% hit rate, our method requires only ca. $10^{10}$ molecules, which is, compared to nano-crystallography, smaller by a factor of 10 ($10^5$ nano-crystals with $10^6\,\mathrm{nm}^3$ volume)[13].

Even higher resolutions are conceivable for larger molecules due to the larger scattering signal[24], albeit computational resources may become a limiting factor when determining larger structures at the same resolution of around 3 Å. However, as shown for the structure determination of the much larger coliphage virus in Supplementary Note 10, the computational complexity only depends on the ratio between the size of the molecule and the desired resolution. For a given resolution, the computational complexity scales slightly faster than the molecular weight cubed.

Given that currently available de novo refinement methods require at least 100 photons per image, we consider our finding that only three photons per image suffice quite unexpected. Further, in this extreme Poisson regime, our three-photon correlation approach—in contrast to previous structure determination methods—allows to compensate for fewer photons per image $P$ by acquiring more images $I$. In particular, because two photons per image do not uniquely determine the structure[39], here we have reached the fundamental limit.

Our analysis also suggests that the method is robust against noise from incoherent scattering, and that removing as much as possible disordered water (or other contaminants) from the molecule in the experiment is crucial. Further, fluctuations of the beam intensity—both in time and due to beam-particle impact parameter fluctuations, which are a limiting factor for image-wise orientation-based methods, should not deteriorate the resolution in our approach, as the correlations are insensitive to such fluctuations. Clearly, further experimental data and improved noise models are required to study the effect of these and other potential noise sources such as background radiation from the evaporated water and detector noise. Structural fluctuations and inhomogeneities of the sample turn more and more into a limiting factor for all current structure determination methods—particularly for high resolutions. Notably, for mixtures of several structures, single-particle scattering implies that the three-photon correlation on which our method rests is a linear superposition of the three-photon correlations of the individual structures. Hence, our approach should be generalizable in a straight-forward way to refine such mixtures, albeit at the cost of more required images, larger computational effort, and more severe convergence issues. Further, due to the averaging properties of the three-photon correlations, our method should be more robust than methods that rely on an accurate orientation of individual scattering images.

We have tested our approach for a conservative estimate of 10 coherently scattered photons. Should the number of coherently scattered photons per shot be larger, e.g., by reducing the size of the beam focus, our method might even bring single-molecule structure determination within reach of less bright free electron lasers or even table top setups[56].

Overall, our results suggest that near-atomic structure determination by single-molecule X-ray scattering is within experimental reach. We would like to point out that our correlation-based method can also determine structures from images containing more than one particle which may further reduce the data acquisition time and facilitate sample delivery (Supplementary Note 11 discusses how the two-photon and three-photon correlation of single molecules is calculated from multi-particle correlations). The method is potentially also useful to extract as much as possible information from other types of scattering experiments, in particular when 3D structures are inferred from noisy two-dimensional projections, such as cryo-EM[57,58], X-ray microscopy, sub-diffractive optical microscopy[59,60], and from fluctuations in correlated X-ray scattering.

## Methods

**Three-photon correlations expressed in SH**. The three-photon correlation $t(k_1, k_2, k_3, \alpha, \beta)$ is the orientational average $\langle \rangle_\omega$ of the product between three intensities $I(\mathbf{k})$ that lie on the intersection between the Ewald sphere and the 3D Fourier density (see Supplementary Note 12),

$$t(k_1, k_2, k_3, \alpha, \beta)_{I(\mathbf{k})} = \left\langle I_\omega\big(\mathbf{k}_1^\star(k_1, 0)\big) \cdot I_\omega\big(\mathbf{k}_2^\star(k_2, \alpha)\big) \cdot I_\omega^\star\big(\mathbf{k}_3^\star(k_3, \beta)\big) \right\rangle_\omega. \quad (1)$$

Here, without loss of generality, the three vectors $\mathbf{k}_1^\star$, $\mathbf{k}_2^\star$, and $\mathbf{k}_3^\star$, are the projection onto the Ewald sphere of the three photons $\mathbf{k}_1 = (k_1, 0, 0)$, $\mathbf{k}_2 = k_2(\cos \alpha, \sin \alpha, 0)$, and $\mathbf{k}_3 = k_3(\cos \beta, \sin \beta, 0)$ in the detector plane. Using a shell-wise SH decomposition of the intensity[47],

$$I(\mathbf{k}) = \sum_{lm} A_{lm}(k) Y_{lm}(\theta, \varphi), \quad (2)$$

with the coefficients $A_{lm}(k)$ describing the intensity function on the respective shells, the three-photon correlation is expressed in sums of products of SH coefficients together with known Wigner-3j symbols and SH basis functions $Y_{lm}(\theta, \varphi)$,

$$\begin{aligned}
t(k_1, k_2, k_3, \alpha, \beta)_{\{A_{lm}(k)\}} = & \sum_{l_1 l_2 l_3} \sum_{m_1 m_2 m_3} A_{l_1 m_1}(k_1) A_{l_2 m_2}(k_2) A_{l_3 m_3}^*(k_3) \\
& \times \begin{pmatrix} l_1 & l_2 & l_3 \\ m_1 & m_2 & -m_3 \end{pmatrix} \sum_{m_1' m_2' m_3'} (-1)^{m_3 - m_3'} \begin{pmatrix} l_1 & l_2 & l_3 \\ m_1' & m_2' & -m_3' \end{pmatrix} \\
& \times Y_{l_1 m_1'}(\theta_1(k_1), 0) Y_{l_2 m_2'}(\theta_2(k_2), \alpha) Y_{l_3 m_3'}^*(\theta_3(k_3), \beta).
\end{aligned} \quad (3)$$

See Supplementary Note 1 for the full derivation of Eq. (3).

**Synthetic data generation**. We validated our structure determination approach using synthetic scattering experiments on the structure of the 46 residue protein Crambin (PDB descriptor: 3U7T)[49] which has been determined to 0.8 Å resolution. To this end, we approximated the 3D electron density $\rho(\mathbf{x})$ by a sum of Gaussian functions centered at the atomic positions with height $\gamma$ and variance $\sigma$ depending on the atom type. The absolute square of the electron densities' Fourier transformation $I(\mathbf{k}) = |\mathcal{FT}[\rho(\mathbf{x})]|^2$ was used to generate synthetic scattering images. In each synthetic scattering experiment, the molecule, and thus also $I(\mathbf{k})$, was randomly oriented. On average $P$ photons per image were generated each shot,

according to the distribution given by the randomly oriented Ewald slice of the intensity $I_\omega(\mathbf{K})$.

To generate the distributions numerically, first, a random set of $N_{pos}$ positions $\{\mathbf{K}_i\}$ in the $k_x k_y$-plane was generated according to a 2D Gaussian distribution $G(\mathbf{K})$ with width $\sigma = 1.05\,\text{Å}^{-1}$. Given a random 3D rotation $\mathbf{U}$ (see Supplementary Note 4 for uniform sampling of SO(3)), rejection sampling method was used to accept or reject each position according to $\xi < I_\omega(\mathbf{U} \cdot \mathbf{K}_i)/(M \cdot G(\mathbf{K}_i))$ using uniformly distributed random numbers $\xi \in [0, 1]$ each. Here, the constant $M$ was chosen as $I_{max} \cdot \max(G(\mathbf{K}))$ such that the ratio $I_\omega(\mathbf{U} \cdot \mathbf{K}_i)/(M \cdot G(\mathbf{K}_i))$ is below 1 for all $\mathbf{K}$. In accordance with our most conservative estimate discussed in the main text, the number of positions $N_{pos}$ was chosen such that on average 10 scattered photons were generated. For assessing the dependency of the resolution on the number of scattered photons, additional image sets with 25, 50, or 100 scattered photons were also generated (Supplementary Note 8).

For technical reasons, we used a SH expansion of the intensity with a high expansion order $L = 35$ as a sufficiently accurate approximation for $I(\mathbf{k})$ to generate the images. The accuracy of the intensity model was cross-checked with the intensity calculated on a cubic grid (150 grid size) using the Fast Fourier Transform, resulting in a 0.9999 correlation, thus establishing sufficient accuracy. Altogether, up to $3.3 \times 10^{10}$ images were generated using a high degree of parallelism.

**Probability of observing a set of triplets.** Because we were not able to derive an analytic inversion for Eq. (3), we chose a probabilistic approach and asked which intensity $I(\mathbf{k})$ is most likely to have generated the complete set of measured scattering images and triplets, respectively. To this end, we considered the probability $p$ that a given intensity $I(\mathbf{k})$, expressed in SH by $\{A_{lm}(k)\}$, generated the set of triplets, $\{k_1^i, k_2^i, k_3^i, \alpha^i, \beta^i\}_{i=1\ldots T}$,

$$p\left(\{k_1^i, k_2^i, k_3^i, \alpha^i, \beta^i\}_{i=1\ldots T} | \{A_{lm}(k)\}\right) = \prod_{i=1}^{T} \tilde{t}(k_1^i, k_2^i, k_3^i, \alpha^i, \beta^i)_{\{A_{lm}(k)\}}. \quad (4)$$

Due to the statistical independence of the triplets, this probability $p$ is a product over the probabilities $\tilde{t}(k_1^i, k_2^i, k_3^i, \alpha^i, \beta^i)$ of observing the individual triplets $i$ which is given by the normalized three-photon correlation $\tilde{t}(k_1, k_2, k_3, \alpha, \beta)$. Here, $\tilde{t}(k_1, k_2, k_3, \alpha, \beta)$ was calculated using Eq. (3) for varying intensity coefficients $\{A_{lm}(k)\}$ and the coefficients that maximized $p(\{k_1^i, k_2^i, k_3^i, \alpha^i, \beta^i\})$ were determined using a Monte Carlo scheme.

In contrast to the direct inversion, the probabilistic approach has the benefit of fully accounting for the Poissonian shot noise implied by the limited number of photon triplets that are extracted from the given scattering images. We note that this approach also circumvents the limitation faced by Kam[39], where only triplets with two photons recorded at the same position could be considered. Because all other triplets had to be discarded, Kam's approach is limited to very high beam intensities, and cannot be applied in the present extreme Poisson regime.

**Reduction of the search space using two-photon correlations.** In our approach, we used the structural information contained within the two-photon correlation to reduce the high-dimensional search space. In analogy to the three-photon correlation, the two-photon correlation is expressed as a sum over products of SH coefficients $A_{lm}(k)$ weighted with Legendre polynomials $P_l$[35,39],

$$c_{k_1, k_2, \alpha} = \sum_l P_l(\cos(\alpha^\star)) \sum_m A_{lm}(k_1)(\omega) A_{lm}^*(k_2). \quad (5)$$

Please note that the $\alpha$ which is seen on the detector is different from the angle $\alpha^\star = \cos^{-1}(\sin(\theta_1)\sin(\theta_2)\cos(\alpha) + \cos(\theta_1)\cos(\theta_2))$ between the two points in 3D intensity space due to the Ewald curvature ($\theta = \cos^{-1}(k\lambda/4\pi)$).

The inversion of Eq. (5) yields coefficient vectors $\mathbf{A}_l^0(k) = (A_{l-m}^0, \ldots, A_{lm}^0)$ for all $l \le L \le K_{max}/2$ and $-l < m < l$, as first demonstrated by Kam[39]. However, all rotations in the $2l + 1$-dimensional coefficient eigenspaces of $\mathbf{A}_l^0(k)$ by $\mathbf{U}_l$ are also solutions,

$$\mathbf{A}_l(k) = \mathbf{U}_l \mathbf{A}_l^0(k). \quad (6)$$

The result implies that the inversion only gives a degenerate solution for the coefficients and the intensity cannot be determined solely from two photons. Here, we used Eq. (6) to search for the optimal rotations $\mathbf{U}_l$ instead of optimal coefficients $A_{lm}^{all}(k)$, which reduced the size of the search space from $\left(\frac{1}{2}L^2 + \frac{3}{2}L + 1\right) \cdot K$ to $\frac{1}{3}\left(L^3 + \frac{15}{4}L^2 + \frac{7}{2}L\right)$ unknowns (e.g., reducing the number of unknowns from 4940 coefficients to 2370 rotation angles for $L = 18$ and $K = 26$). See Supplementary Note 6 for more details.

**Monte Carlo simulated annealing.** The probability $p$ from Eq. (4) was maximized by a Monte Carlo/simulated annealing approach on the energy function:

$$E\left(\{k_1^i, k_2^i, k_3^i, \alpha^i, \beta^i\} | \{A_{lm}(k)\}\right) = -\log p\left(\{k_1^i, k_2^i, k_3^i, \alpha^i, \beta^i\} | \{A_{lm}(k)\}\right)$$
$$= -\sum_i \log \tilde{t}(k_1^i, k_2^i, k_3^i, \alpha^i, \beta^i)_{\{A_{lm}(k)\}}, \quad (7)$$

in the space of all rotations $\mathbf{U}_l$ given by the inversion of the two-photon correlation. Each Monte Carlo run was initialized with a random set of rotations $\{\mathbf{U}_l\}$ and the set of unaligned coefficients $\{\mathbf{A}_l^0\}$. In each Monte Carlo step $j$, all rotations $\mathbf{U}_l^j$ were varied by small random rotations $\Delta_l(\beta_l)$ such that the updated rotations for each $l$ ($l \le L$) read $\mathbf{U}_l^{j+1} = \Delta_l(\beta_l) \cdot \mathbf{U}_l^j$ using stepsizes $\beta_l$. In order to escape local minima, a simulated annealing was performed using an exponentially decaying temperature protocol, $T(j) = T_{init}\exp(j/\tau)$. Steps with an increased energy were also accepted according to the Boltzmann factor $\exp(-\Delta E/T)$. We further used adaptive stepsizes such that all $\beta(l)$ were increased or decreased by a factor $\mu$ when accepting or rejecting the proposed steps, respectively. Convergence was improved by using a hierarchical approach in which the intensity was first determined with low angular resolution and further increased to high resolution. To this end, the variations of low-resolution features were frozen out faster than the variations of high-resolution features. See Supplementary Note 4 on how to generate random rotations in SO(n) and how the parameters of the Monte Carlo search were determined.

**Calculation of real space electron densities and resolutions.** Supplementary Fig. 7 summarizes the calculation of the electron densities as carried out in this work. All intensities were obtained up to an arbitrary Euler rotation ($\theta, \phi, \psi$) and were therefore rotationally fit to the known reference intensity for subsequent comparison. The phases of the aligned intensities were calculated using the relaxed averaged alternating reflections (RAAR) method by Luke[45]. The resolution of the electron densities was characterized by the FSC,

$$\text{FSC}(k) = \frac{\sum_{k_i \in k} F_1(k_i) \cdot F_2(k_i)^*}{\sqrt[2]{\sum_{k_i \in k} |F_1(k_i)|^2 \cdot \sum_{k_i \in k} |F_2(k_i)|^2}}. \quad (8)$$

In analogy to cryo-EM[51], the resolution is defined as the wave number $k_{res}$ at which FSC($k$) = 0.5, yielding a radial resolution $\Delta r = 2\pi/k_{res}$.

Starting from an individual set of doublet and triplet histograms (Supplementary Fig. 1), 20 independent intensity determination runs were carried out to asses and improve convergence of the Monte Carlo simulated annealing runs. To reduce the phasing error, the phase retrieval of one intensity was carried out eight times and the resulting eight electron densities were averaged. The final electron density, for which the resolution is given, is the average of those 20 individual densities and the resolution error was estimated from the standard deviation of the resolution of the 20 individual electron densities. We chose to average in real space instead of Fourier space before phasing because we found that this sequence yielded more accurate electron densities.

**Evaluation of phasing errors.** To asses the phasing error, we compared the intensities of the phased electron densities $I_{phased} = |\mathcal{FT}[\rho_{retrieved}]|^2$ with the intensities $I_{retrieved}$ before phasing. To this end, the ISC was calculated as:

$$\text{ISC}(k) = \frac{\sum_{k_i \in k} \left(I_{res}(k_i) - \overline{I_{res}(k_i)}\right)\left(I_{ref}(k_i) - \overline{I_{ref}(k_i)}\right)}{\sqrt{\sum_{k_i \in k} \left(I_{res}(k_i) - \overline{I_{res}(k_i)}\right)^2}\sqrt{\sum_{k_i \in k} \left(I_{ref}(k_i) - \overline{I_{ref}(k_i)}\right)^2}}. \quad (9)$$

In analogy to the FSC, we considered ISC($k$) = 0.5 as a resolution measure. As can be seen in Supplementary Fig. 8, the phasing shifted this crossover from approx. 2.8 to 3.1 Å, but does not distort the shapes and relative heights of the ISC curves. Assuming that the phasing error can be estimated from the shift of this crossover, for our high-resolution density result with 3.3 Å resolution (retrieved from $3.3 \times 10^{10}$ photons), a decrease in resolution of ca. 0.3 Å is expected to be due to phasing.

**Data availability.** All relevant data are available from the authors.

**Code availability.** The code is available at https://github.com/h4rm/ThreePhotons.jl and the data analysis was done using IJulia notebooks which are available at https://github.com/h4rm/ThreePhotonsNotebook. For more information, please visit http://www.mpibpc.mpg.de/grubmueller/threephotons.

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

## Acknowledgements

Financial support from the Deutsche Forschungsgemeinschaft (DFG) Grant No. SFB 755.B4 and helpful discussions with Russel Luke are gratefully acknowledged.

## Author contributions

B.v.A., M.M., H.G. conceived research, B.v.A. carried out research, B.v.A., M.M., H.G. wrote paper.

## Additional information

**Competing interests:** The authors declare no competing interests.

