## [Peer Review File · Nature Communications]

Reviewers' comments:

Reviewer #1 (Remarks to the Author):

The manuscript describes a new method for using three photon correlations to obtain the structure of single molecules from X-ray scattering experiments.

I find the paper very interesting and I think the method might be a viable new way of analysing single molecule diffraction data. The results of the simulations, using relatively realistic parameters, demonstrate resolution better than 3.5 Å, which is very impressive. One might argue that there are several important noise contributions which were not taken into account, such as scattering from beamline components or dynamics of the sample itself, but this leads to an endless search for the perfect simulations and I think the scope of the simulations presented in the manuscript are clearly enough to warrant publication.

There are however several points that require clarifications or corrections:

- In the abstract you say that you can achieve the results with 3.3×10^{10} recorded photons. I think it would be good to clarify these are elastically scattered photons, and somehow allude to the fact that these are collected over a large number of shots.
- Page 2 "allow for extremely high peak brilliance...". I think there's a "pulses" missing after brilliance.
- Page 2, end of first paragraph. You state the number of scattered photons but do not specify the wavelength. Also there are experimentally available numbers on photon flux at the interaction region from:
 1. Hantke, M. F. et al. High-throughput imaging of heterogeneous cell organelles with an X-ray laser. *Nat. Photonics* 8, 943–949 (2014).
 2. Daurer, B. J. et al. Experimental strategies for imaging bioparticles with femtosecond hard X-ray pulses. *IUCrJ* 4, 251–262 (2017).
- Page 2, when you reference orientation determination methods "[17-22]", I think some of the work from G. Bortel, e.g. "Tegze, M. & Bortel, G. Atomic structure of a single large biomolecule from diffraction patterns of random orientations. *J. Struct. Biol.* 179, 41–5 (2012)." deserves a mention.
- Page 3 "determination was reported for much more". Probably should be "determination was only reported for much more".
- Page 4. The sentence before section II is a bit too strong, as this is a purely theoretical work. There are many challenges, such as sample delivery, on the path of the successful experiment.
- Page 4 "approximately planar intersection". Given the wavelengths used and resolutions aimed for the Ewald sphere is far from planar.
- Just a bit below "which is the absolute square" -> "which is proportional to the absolute square"
- Page 5. You swapped the cross section of N and O. More importantly there seems to be a misunderstanding on calculating the total scattering cross section for Crambin. You seem to have summed the cross sections of the atoms, but as this is a coherent process the interference terms between the atoms have to be taken into account. This results in a total coherent scattering cross section much higher than the one you stated. For example with the parameters you describe I calculated about 600 scattered photons, instead of the 20 you estimate. Check e.g. Hantke, M. F., Ekeberg, T. & Maia, F. R. N. C. Condor: A simulation tool for flash X-ray imaging. *J. Appl. Crystallogr.* 49, 1356–1362 (2016) on how to estimate the coherent scattering.
- Page 5. The assumption of a 10% hit-rate is quite optimistic. Most experiments run closer to 1%. Also a few days of data collection is more than any experiment gets. While beamtime might last up to 60 hours, data collection time rarely goes over 24h.
- It would be good to indicate somewhere that your convention for reciprocal space uses $k_{in} = 2\pi/\lambda$ and not $k_{in} = 1/\lambda$, which is commonly used in most other papers.

- Page 8. The "gold standard" refers to not doing any operations to the data before dividing it into two parts and doing independent reconstructions. It does not refer to any particular cutoff values.
- Page 8 you should make it more explicit between what did you calculate the FSC. I imagine it's between the averaged electron density and the "solution".
- Page 11. You overestimate the amount of incoherent scattering (because you underestimate the coherent). For incoherent the cross section is simply a sum of the atomic cross sections.
- Page 12 "For scattering at disordered" -> "For scattering from disordered"
- Page 12, again 10% hit rate is **not** conservative.
- Top of Page 13 This comparison between single particle and nanocrystallography only accounts for the molecules that see the beam totally ignoring the efficiencies of sample delivery and so it's not experimentally relevant.
- Page 13, when you mention other potential noise sources I think an important one that should be stated is the fluctuations of the structure of the sample itself!

The following comments refer to the supplementary material:

- Page 2, the planar approximation is wrong for 5 keV and 3.3 Å. The the maximum angle scattered is more than 40 degrees!
- Page 3, you state Friedel's law but apply it to intensity. As intensity is real the complex conjugate does not make much sense.
- Throughout the text you use the dot "." both as decimal and thousands separator. Please fix it.
- Page 18. There's an extra "described in Sec. V."

I would like to salute the authors for making the code used in this work publicly available online. I think this is a crucial part of any such complex simulation work and should be made mandatory, as it provides an objective description of the steps taken and greatly assists reproducibility.

Overall I think this is a good paper but the above issue need to be corrected before it can be published.

Reviewer #2 (Remarks to the Author):

This paper describes an approach to recovering three-dimensional structure from single-biomolecule X-ray diffraction patterns. The approach involves Bayesian analysis of three-point correlations from diffraction patterns containing as few as three photons. Validation is performed with synthetic data with Poisson noise, with and without background scattering.

The Bayesian approach to recovering structure from three-point correlations in low-signal diffraction patterns is interesting, and creatively implemented.

General Comments

1. There are perhaps a dozen different methods able to extract structure from synthetic diffraction patterns, in some cases at extremely low photon counts. Validation with experimental data is now indispensable for a new approach to command attention.
2. A key challenge in analyzing experimental single-particle XFEL data stems from the need to deal with multiple types of inhomogeneity. These include variations in incident intensity, extraneous scattering, beam-particle impact parameter, detector response, and particle structure. It is important to demonstrate the ability of the proposed approach to extract reliable information from inhomogeneous data.
3. Well-characterized experimental single-particle XFEL data are now publicly available [see: Reddy et al., *Sci Data* 4, 170079 (2017)]. Validation of the proposed approach with such experimental data

would substantially enhance its impact.

Specific Comments

1. Can the authors address the uniqueness of the solution obtained by their Bayesian search?
2. The key parameters for characterizing a single-particle diffraction pattern are the number of scattered photons per speckle, and the resolution. The number of scattered photons per diffraction pattern is inadequate, because it ignores the computational complexity of the problem at hand. The Conclusions section notes, in passing, that computational resources may become a limiting factor for particles larger than the (very small) protein considered in the paper. I recommend computational expense be discussed in greater detail. What is the largest object, which can be treated by the proposed approach?
3. The assumed hit-rate of 10% is optimistic. Current hit-rates are typically two orders of magnitude lower, with few ideas for substantial improvement. Would this preclude the practical use of the approach? What problems could be tackled at current hit-rates?
4. The Gaussian model used for simulating background scattering is unrealistic. Experimental background scattering is structured, and changes from diffraction pattern to diffraction pattern.

Reviewer #3 (Remarks to the Author):

Von Ardenne, Mechelke and Grubmüller discuss the possibilities and limitations of XFEL solution scattering experiments and analysis for the case of a small protein (crambin, ca. 4.7 kDa), using full three-photon correlation as an orientation-independent representation of the scattering images. Since the analytic expression of the three-photon correlation could not be inverted and the number of unknowns is too large for a numeric solution, the authors relied on a probabilistic approach and solved for the spherical harmonics coefficients that maximize the probability of observing all recorded triplets. Importantly, it appears that smaller molecules constitute more challenging candidates for 3D-structure determination by scattering than large complexes such as viral particles. The presented work contends that it may be possible to reach a resolution of 3.3 Å for a small protein with realistic beam times (days rather than weeks) and photon counts per pulse in a range obtainable at XFEL setups currently in operation or under construction. For comparison, the X-ray single crystal structure of crambin has been determined at resolutions of up to 0.5 Å and diffraction data collection can be accomplished in a few minutes.

Crystal-free 3D structure determination at resolutions that allow visualization of H-bonds remains a dream, albeit one that may one day become reality. The authors provide a theoretical framework that is consistent with the possibility of obtaining resolutions for a small-protein structure that would allow tracing of the backbone and perhaps make visible some of the side chains. XFEL scattering could thus complement single crystal crystallography and cryo-EM and become competitive with NMR in terms of the time needed for data collection and peak assignment (large proteins have remained off-limits to NMR). In this vein, the present manuscript will certainly elicit excitement by many structural biologists.

The authors discuss various experimental challenges that could stand in the way of reaching the structural resolution indicated by the theoretical treatment. These include incoherent scattering, background radiation, detector noise, and scattering at the unstructured fraction of water molecules that may adhere to the surface of the macromolecules. In that sense experimental realities have a way of casting a shadow on theoretical predictions. However, the authors make at least an effort to take these into account and estimate the loss in resolution for various scenarios. A rather mundane but important limitation is the fact that access to XFEL setups is currently very limited. Moreover, the technical hurdles (either nanocrystal generation or solution experiments, data processing, etc.) are

most likely too high to be overcome by the large majority of labs and researchers not intimately associated with one of the setups in the US or Europe. Concentration-dependent protein aggregation is a common pitfall of scattering experiments (i.e. SAXS/SANS; not necessarily of successful crystallizations though), and a protein/protein construct that suffers from it will be doomed in terms of structure determination by XFEL scattering as well. Theoretical predictions can sometimes be too optimistic in terms of the experimental possibilities: phosphorus SAD has run into a wall but sulfur phasing has worked out beyond most expectations in crystallographic phasing; neutron diffraction has not lived up to its promises even as higher flux SNS beam lines have come on line and crystal size remains a limitation; etc.

Nevertheless, despite these concerns, the present manuscript reporting the potential benefits and future possibilities of XFEL scattering for structural biology are pretty exciting and worth publishing.

Minor corrections:

- 1) Page 11, 4 lines from bottom: which corresponds.
- 2) Page 13, 5 lines from end of text: relevant (relevant).

Reviewer #1:

The manuscript describes a new method for using three photon correlations to obtain the structure of single molecules from X-ray scattering experiments. I find the paper very interesting and I think the method might be a viable new way of analysing single molecule diffraction data. The results of the simulations, using relatively realistic parameters, demonstrate resolution better than 3.5 Å, which is very impressive.

We thank the Referee for this clear and very positive assessment.

One might argue that there are several important noise contributions which were not taken into account, such as scattering from beamline components or dynamics of the sample itself, but this leads to an endless search for the perfect simulations and I think the scope of the simulations presented in the manuscript are clearly enough to warrant publication.

We fully agree -- and add that, because of the current lack of small molecule experiments, some of the noise contributions are simply still not sufficiently characterized to enable inclusion into our simulations. By comparison, we do expect our method, as demonstrated in the manuscript, to be more robust with other published methods against, shot noise, fluctuations of beam intensity or background noise due to scattering from beamline components or residual gas and/or evaporated solvents.

Further, also by construction and in contrast to many other methods, our method should be extendable to also include structural heterogeneity of the sample, which however is also beyond the scope of this manuscript.

We have changed parts of the discussion (p. 16-17) to explain and justify this point more clearly.

- In the abstract you say that you can achieve the results with $3.3e10$ recorded photons. I think it would be good to clarify these are elastically scattered photons, and somehow allude to the fact that these are collected over a large number of shots.

We fully agree and have **now clarified these two points in the abstract.**

- Page 2 "allow for extremely high peak brilliance...". I think there's a "pulses" missing after brilliance.

Thanks, added.

- Page 2, end of first paragraph. You state the number of scattered photons but do not specify the wavelength. Also there are experimentally available numbers on photon flux at the interaction region from:

1. Hantke, M. F. et al. High-throughput imaging of heterogeneous cell organelles with an X-ray laser. Nat. Photonics 8, 943–949 (2014).

2. Daurer, B. J. et al. *Experimental strategies for imaging bioparticles with femtosecond hard X-ray pulses. IUCrJ* 4, 251–262 (2017).

As stated in more detail below, we now account for the Ewald curvature and have added the wavelength (2.5 Å) where requested. For the photon flux we have used the estimate for the European XFEL by

Yoon, C. H., Yurkov, M. V., Schneidmiller, E. A., Samoylova, L., Buzmakov, A., Jurek, Z., ... Mancuso, A. P. (2016). **A comprehensive simulation framework for imaging single particles and biomolecules at the European X-ray Free-Electron Laser.** *Scientific Reports*, 6(24791), 24791.
<https://doi.org/10.1038/srep24791>

which uses 5×10^{11} photons per beam and assumed a beam diameter of 100 nm, resulting in the 6.3×10^7 photons/nm² flux.

In their experiments in 2014, Hantke et. al. used a beam with 2.2×10^{12} photons per pulse focused to 5 μm resulting in flux of 1.1×10^6 photons/nm².

Daurer also estimated the peak beam intensity with 1.9×10^6 photons /nm². Both measurements were taken at the (not upgraded) LCLS and we think they mark a lower bound for the expected flux at next-generation XFELs.

Other methods, such as the one from Tegze (2012) use 10^{12} photons for 100 nm beam diameter (1.2×10^8 photons/nm² flux) and we therefore consider our estimate for the photon flux to be well on the conservative side.

- Page 2, when you reference orientation determination methods "[17-22]", I think some of the work from G. Bortel, e.g. "Tegze, M. & Bortel, G. Atomic structure of a single large biomolecule from diffraction patterns of random orientations. *J. Struct. Biol.* 179, 41–5 (2012)." deserves a mention.

We thank the Reviewer for pointing us to this work, which we now mention in the introduction (p. 3).

- Page 3 "determination was reported for much more". Probably should be "determination was only reported for much more".

Thanks, corrected.

- Page 4. The sentence before section II is a bit too strong, as this is a purely theoretical work. There are many challenges, such as sample delivery, on the path of the successful experiment.

We have now changed the sentence into: "[...], such that near-atomic resolution for single biomolecules should in principle be possible even at such extremely low photon counts."

- Page 4 "approximately planar intersection". Given the wavelengths used and resolutions aimed for the Ewald sphere is far from planar.

We fully agree with the Referee and now **have extended our synthetic experiments and the structure determination method to work with variable wavelengths and Ewald curvatures, respectively.**

As a test case, we have repeated the complete structure determination protocol with realistic wavelengths at 2.5 Å resolution for 10^9 and 10^8 images using the exact Ewald sphere. As expected, the achieved resolution and structure accuracy did not change compared to the image set generated and analyzed using the previous planar intersection (equivalent to 0 Å wavelength), such that also our conclusions are unaffected. **We have now included the full Ewald sphere derivation in the text and point out that the planar intersection was used for the synthetic experiments, and refer to the above control in the Supplement.**

- *Just a bit below "which is the absolute square" -> "which is proportional to the absolute square"*

Corrected, thanks.

- *Page 5. You swapped the cross section of N and O. More importantly there seems to be a misunderstanding on calculating the total scattering cross section for Crambin. You seem to have summed the cross sections of the atoms, but as this is a coherent process the interference terms between the atoms have to be taken into account. This results in a total coherent scattering cross section much higher than the one you stated. For example with the parameters you describe I calculated about 600 scattered photons, instead of the 20 you estimate. Check e.g. Hantke, M. F., Ekeberg, T. & Maia, F. R. N. C. Condor: A simulation tool for flash X-ray imaging. *J. Appl. Crystallogr.* 49, 1356–1362 (2016) on how to estimate the coherent scattering.*

Thanks for catching the N/O swap - corrected.

We also thank the Referee to point out this issue and confirm that the Condor tool by Maia, Ekeberg and Hantke predicts up to $1e3$ coherently scattered photons for our Crambin molecule.

However, the SIMEX tool by Fortmann–Grote *et al.* from DESY, for which we were provided a realistic XFEL beam profiles from the authors of the framework, estimated only 16 scattered photons which corresponds to our very simple estimate involving summing up the coherent scattering cross sections.

Unfortunately we could not resolve the contradictory results yet and **have now stated these two numbers in the paper on p.6, with 16 scattered photons being the most conservative** -- and more challenging -- estimate. As also discussed in the paper, more photons per picture would drastically reduce the required beam time, so we think we are on the safe side here.

Further, as we also have now mentioned in our revised manuscript, higher number of photons than expected would imply that our method would also allow single molecule experiments using less bright FEL sources such as the SwissFEL or SACLA (or even table top experiments), which would further expand the scope of our approach.

Hence, whatever the experiments will tell, our conclusions do not depend on this issue. **A commentary sentence has been added to the Conclusion on p.16/17.**

- Page 5. The assumption of a 10% hit-rate is quite optimistic. Most experiments run closer to 1%. Also a few day of data collection is more than any experiment gets. While beamtime might last up to 60 hours, data collection time rarely goes over 24h.

We have now updated our estimates and include both 1% and 10% hit rate as conservative estimate and as an estimate that shows the potential of increasing the hit rate. However, as one additional advantage of our method that was not discussed in the manuscript we now mention in the revised manuscript that, by the construction as a correlation based method, also those events with two or more molecules in the beam focus add valuable structural information to our signal. We are certainly not experts in the experimental details, but would assume that the main limiting factor for the hit rate is the need to keep the probability of multiple scattering events low. Because our methods should enable to relax this requirement, higher particle concentrations and correspondingly higher hit rates should be possible. Concerning the maximum available data collection time of 24h, we would argue that the best we can do is to provide all the information required to determine the achievable resolution for given data collection time, focus size, hit rate etc. – which is what we did.

- It would be good to indicate somewhere that your convention for reciprocal space uses $k_{in} = 2\pi/\lambda$ and not $k_{in} = 1/\lambda$, which is commonly used in most other papers.

We have now pointed out this convention in our text (p.10).

- Page 8. The "gold standard" refers to not doing any operations to the data before dividing it into two parts and doing independent reconstructions. It does not refer to any particular cutoff values.

We have changed the text now to avoid this possible misunderstanding (p.10).

- Page 8 you should make it more explicit between what did you calculate the FSC. I imagine it's between the averaged electron density and the "solution".

As the Reviewer supposed, the FSC curves shown in Fig.3 on p.8 are indeed between the averaged electron density and the known electron density. We have now updated both the text and the caption to make this clear.

- Page 11. You overestimate the amount of incoherent scattering (because you underestimate the coherent). For incoherent the cross section is simply a sum of the atomic cross sections.

If, in fact, we have underestimated the the coherent scattering signal, then this is absolutely right. See discussion above.

- Page 12 "*For scattering at disordered*" -> "*For scattering from disordered*"

Corrected, Thank you.

- Page 12, again 10% hit rate is **not** conservative.

We have also changed this instance as described above, and have removed that claim that a 10% hit rate is conservative.

- Top of Page 13 *This comparison between single particle and nanocrystallography only accounts for the molecules that see the beam totally ignoring the efficiencies of sample delivery and so it's not experimentally relevant.*

For our estimate we have assumed the above 10% hit rate, which we understand as part of the efficiency of sample delivery. We agree that this estimate indeed ignored other sample losses, most importantly the fact that crystallization can only proceed in slightly over-saturated solution and stops as soon as the protein concentration drops below. A back-of-the-envelope calculation with typical values (average sized protein of 10 kD and 3 nm diameter, 1000 microcrystals from 1 microliter protein solution, 5mg/ml protein concentration) yields 3×10^{14} proteins in solution and of which only 3×10^{10} end up in the 1000 crystals – the rest remains in solution and is typically lost. In other words: only one out of 10000 proteins makes it into the crystal. This loss would not occur in single molecule scattering experiments. So even if one allows for other losses such as a reduced hit-rate, we think it is safe to say that single molecule scattering experiments require less material.

- Page 13, when you mention other potential noise sources I think an important one that should be stated is the fluctuations of the structure of the sample itself!

We fully agree, and in fact the structural fluctuations of the sample turn more and more into a limiting factor for *all* structure determination methods (X-ray crystallography, NMR, and also cryo electron microscopy) --- particularly for resolutions higher than 2.5 Å. At 3-4 Å, thermal fluctuations do not pose a significant challenge though. Larger, collective structural dynamics, do pose a challenge to all methods. Clearly, the strategy of sorting the raw images into structure classes, as routinely done in cryo-EM, is not an option here. However, our Bayes-approach can be extended in a straight-forward way to simultaneously refine mixtures of a few structures, albeit at the cost of more required images, larger computational effort, and more severe convergence issues. We have therefore not yet attempted to solve this issue, and are unaware of any other method for single molecule X-ray scattering that actually has. **As suggested, we now mention structure fluctuations in our revised manuscript, and have added a short summary of the above discussion (p.16/17).**

The following comments refer to the supplementary material:

- Page 2, the planar approximation is wrong for 5 keV and 3.3 Å. The the maximum angle scattered is more than 40 degrees!

We have now included the Ewald curvature, as stated above, and have corrected all mentions of the planar approximations, respectively. As is now shown in the Supplement for test cases, all results and conclusions are essentially unaffected.

- Page 3, you state Friedel's law but apply it to intensity. As intensity is real the complex conjugate does not make much sense.

Thank you for pointing out this notational error, **which we have now corrected**.

- Throughout the text you use the dot "." both as decimal and thousands separator. Please fix it.

Fixed – German disease.

- Page 18. There's an extra "described in Sec. V."

Corrected.

I would like to salute the authors for making the code used in this work publicly available online. I think this is a crucial part of any such complex simulation work and should be made mandatory, as it provides an objective description of the steps taken and greatly assists reproducibility.

Thank you, we couldn't agree more.

Reviewer #2:

This paper describes an approach to recovering three-dimensional structure from single-biomolecule X-ray diffraction patterns. The approach involves Bayesian analysis of three-point correlations from diffraction patterns containing as few as three photons. Validation is performed with synthetic data with Poisson noise, with and without background scattering.

The Bayesian approach to recovering structure from three-point correlations in low-signal diffraction patterns is interesting, and creatively implemented.

General Comments

1. *There are perhaps a dozen different methods able to extract structure from synthetic diffraction patterns, in some cases at extremely low photon counts. Validation with experimental data is now indispensable for a new approach to command attention.*

We agree that the development of theory and methods for single molecule X-ray scattering structure extraction is indeed a very active field, and has in the past few years produced a number of complementary approaches to this end. We also understand the strong wish on the side of the experimentalists to know which of these methods *really* works and which not – a question which validations against synthetic data can only partially answer. But we respectfully disagree with the Reviewer's suggestion (unfortunately without references) that many other methods for low photon counts were already available. As discussed in the introduction, our method pushes the limit by a factor of almost 100 as compared to the best published method, which requires several 100 photons per image [Walczak, M., & Grubmüller, H. (2014). Bayesian orientation estimate and structure information from sparse single-molecule x-ray diffraction images. *Physical Review E - Statistical, Nonlinear, and Soft Matter Physics*, 90(2), 22714]. We note that almost all publications report the number of photons **per (Sannon) pixel** rather than the much larger total number of photons per image we're referring to. Also because we have reached the fundamental limit of three photons per image, we think (together with the two other Reviewers) that this is a sufficient advancement that should be communicated and not withheld until measurements have been performed. In fact, we think that our results can also help in the design of the experiments, and motivate groups to push the limits towards smaller samples.

2. *A key challenge in analyzing experimental single-particle XFEL data stems from the need to deal with multiple types of inhomogeneity. These include variations in incident intensity, extraneous scattering, beam-particle impact parameter, detector response, and particle structure. It is important to demonstrate the ability of the proposed approach to extract reliable information from inhomogeneous data.*

We fully agree that inhomogeneities / noise pose a particular challenge, and certainly do not claim to have solved all issues of such a large challenge in one single theory paper – in our view, that would be asking for too much. Fully in line with the comments of both Reviewers #1 and #3, we assessed the impact of those sources of inhomogeneities and noise for which sufficiently accurate data or models are available (or can be calculated), such that a quantitative assessment makes sense. In contrast to what the Reviewers comment might suggest, we therefore **did** include the effect of beam intensity fluctuations, incoherent scattering, extraneous scattering due scattering from solvent molecules adhering to the sample molecule.

We would specifically stress that because our method is correlation based, it is insensitive to variations in incident intensity and beam-particle impact parameter. In the Conclusion (p.13) of the original manuscript, this fact was stated as "[robust against] fluctuations of the beam intensity – both in time and space", which may have been unclear. **We have reformulated this part to enhance clarity and highlight the fact that, to our best knowledge, all other published methods have not overcome this problem.**

3. *Well-characterized experimental single-particle XFEL data are now publicly available [see: Reddy et al., Sci Data 4, 170079 (2017)]. Validation of the proposed approach with such experimental data would substantially enhance its impact.*

As written in the manuscript (p. 15), we would be more than happy to have the opportunity to test our method against low photon count experimental data, and have in fact initiated already a collaboration to that aim. Unfortunately, such data has not been obtained so far. The Reviewer pointed out the data by Reddy *et al.* (published three days before submission of our manuscript), which indeed is a significantly improved data set -- and in fact an impressive experimental achievement --, over the Mimivirus scattering experiments published in 2015 (our ref [8]). Because the 65–70 nm coliphage PR772 particle at ca 10 nm resolution scatters ca. 4×10^5 photons per image, it is still far outside the low photon count regime for which our method was devised and optimized. Nevertheless, and following the suggestion of the Referee, **we have now included a full structure determination from these data set (see Supplement)**, and are happy to report we have successfully obtained a three dimensional density of the virus similar to the one reported in [Hosseinizadeh, A., Mashayekhi, G., Copperman, J., Schwander, P., Dashti, A., Sepehr, R., ... Ourmazd, A. (2017). Conformational landscape of a virus by single-particle X-ray scattering. *Nature Methods*, 14(9), 877–881]. **We have now also added a respective note at the end of the Results & Discussion Section (p.15).**

Specific Comments

1. *Can the authors address the uniqueness of the solution obtained by their Bayesian search?*

The mathematical results on two- and degenerated three photon correlations by Kam, 1980 have already suggested that the solution, i.e., the minimum of our Bayes cost function, should indeed be unique (apart from trivial rotations). This

result is strongly supported in our paper by (1) the observation that repeated optimization runs starting from completely un-correlated starting points always converge to the same Fourier space density and (2) both converged Fourier space density and electron density in real space agree with the known reference structure from which the synthetic data was generated. **We have now added this comment to the methods section (p.6)**

2. *The key parameters for characterizing a single-particle diffraction pattern are the number of scattered photons per speckle, and the resolution. The number of scattered photons per diffraction pattern is inadequate, because it ignores the computational complexity of the problem at hand. The Conclusions section notes, in passing, that computational resources may become a limiting factor for particles larger than the (very small) protein considered in the paper. I recommend computational expense be discussed in greater detail. What is the largest object, which can be treated by the proposed approach?*

Following this suggestion, **we now provide more information on the computational complexity in the conclusion (p.16)**, in addition to our more qualitative remark in the original manuscript that for large systems the required computational effort may become an issue. In reading the Reviewer's comment we think this remark may have also created the (wrong) impression that our methods is unable of dealing with larger particles – quite the contrary, as also pointed out by Reviewer #3, larger particles are easier because they scatter more photons per image. What determines the computational effort is not the size of the sample, but the ratio between size and achieved resolution: e.g., for the coliphage PR772 particle for which we have now also determined the structure with our method (see Supplement) at a resolution of ca 10 nm, the computational effort was similar to our 3 nm crambin protein at 0.35 nm resolution. Of course, the phage at atomistic resolution -- *that* would pose a severe computational challenge and thus require improved optimization algorithms.

3. *The assumed hit-rate of 10% is optimistic. Current hit-rates are typically two orders of magnitude lower, with few ideas for substantial improvement. Would this preclude the practical use of the approach? What problems could be tackled at current hit-rates?*

There seems to be a broader range of 'realistic' hit rates both in the literature, e.g.,

0.1% hit-rate: Donatelli, J. J., Sethian, J. A., & Zwart, P. H. (2017). Reconstruction from limited single-particle diffraction data via simultaneous determination of state, orientation, intensity, and phase. *PNAS*, 201708217

1 - 20 %: Kirian, R. A. (2012). Structure determination through correlated fluctuations in x-ray scattering. *Journal of Physics B: Atomic, Molecular and Optical Physics*, 45(22), 223001

and in the Reviewers' comments (e.g., Referee #1 considers 1% realistic).

Given this current uncertainty, **we have now included within our revised manuscript estimates of what can be achieved for the various discussed hit rates.** As similar uncertainty exists with respect to what focus size is realistic, which we also have discussed in the manuscript. We think all Reviewers will agree that the best we can do is to provide all the information required such that experimentalists can easily determine which resolution can be expected for given beam-time, repetition rate, focus size/intensity, hit rate, and noise level. **We have now modified the text accordingly (p. 7).**

4. *The Gaussian model used for simulating background scattering is unrealistic. Experimental background scattering is structured, and changes from diffraction pattern to diffraction pattern.*

We fully agree, and in fact had written in the original manuscript, that "further experimental data and improved noise models are required". **As we have now discussed more clearly in the conclusion on p.16,** we think that variations and structure in the background scattering is a lesser problem, because – in contrast to most other methods that work a few 100 photons per image, our correlation based method does not rely on any analysis at the level of single images (such as orientation estimate), but averages all images into a single histogram in 3-photon-correlation space. Due to this averaging, variations of background scattering are likely to average out. The main challenge, therefore, is the sheer *amount (fraction of photons)* of background scattering, which is already captured by the simple Gaussian model and which, therefore, we think we have already addressed.

Reviewer #3:

Von Ardenne, Mechelke and Grubmüller discuss the possibilities and limitations of XFEL solution scattering experiments and analysis for the case of a small protein (crambin, ca. 4.7 kDa), using full three-photon correlation as an orientation-independent representation of the scattering images. Since the analytic expression of the three-photon correlation could not be inverted and the number of unknowns is too large for a numeric solution, the authors relied on a probabilistic approach and solved for the spherical harmonics coefficients that maximize the probability of observing all recorded triplets. Importantly, it appears that smaller molecules constitute more challenging candidates for 3D-structure determination by scattering than large complexes such as viral particles. The presented work contends that it may be possible to reach a resolution of 3.3 Å for a small protein with realistic beam times (days rather than weeks) and photon counts per pulse in a range obtainable at XFEL setups currently in operation or under construction. For comparison, the X-ray single crystal structure of crambin has been determined at resolutions of up to 0.5 Å and diffraction data collection can be accomplished in a few minutes.

Crystal-free 3D structure determination at resolutions that allow visualization of H-bonds remains a dream, albeit one that may one day become reality. The authors provide a theoretical framework that is consistent with the possibility of obtaining resolutions for a small-protein structure that would allow tracing of the backbone and perhaps make visible some of the side chains. XFEL scattering could thus complement single crystal crystallography and cryo-EM and become competitive with NMR in terms of the time needed for data collection and peak assignment (large proteins have remained off-limits to NMR). In this vein, the present manuscript will certainly elicit excitement by many structural biologists.

We thank the Reviewer for these comments. We are glad he/she shares our view that, whereas one single paper cannot solve all theoretical and experimental problem in one stroke, it establishes and reaches the fundamental three-photon limit and thus provides new evidence that this dream is indeed a realistic one. Of course, we certainly do not want to compete with X-ray crystallography for Crambin-like proteins that can easily be crystallized, as we are sure the Reviewer also did not want to suggest above. Rather, we chose it as a challenging test because (1) it is small, thus scattering very few photons, and (2) its structure has been determined to very high resolution such that our synthetic scattering experiments are accurate and realistic.

The authors discuss various experimental challenges that could stand in the way of reaching the structural resolution indicted by the theoretical treatment. These include incoherent scattering, background radiation, detector noise, and scattering at the unstructured fraction of water molecules that may adhere to the surface of the macromolecules. In that sense experimental realities have a way of casting a shadow on theoretical predictions. However, the authors make at least an effort to take these into account and estimate the loss in resolution for various scenarios.

We thank the Reviewer for acknowledging our efforts to quantitatively assess the effect of as many different sources of noise in real experiments as currently reasonably possible, also in light of the comments of Reviewer #2 on the same issue. We should like to add that, in fact, we are not aware of any publication developing a new method for structure determination in the low photon count range that would address and/or assess a larger number of different sources of noise or inhomogeneity.

A rather mundane but important limitation is the fact that access to XFEL setups is currently very limited. Moreover, the technical hurdles (either nanocrystal generation or solution experiments, data processing, etc.) are most likely too high to be overcome by the large majority of labs and researchers not intimately associated with one the setups in the US or Europe.

The Reviewer brings up an important issue: Because access to XFELs is rather limited, it is essential that as much structural information as possible is extracted from the experiments – from every single image. Because our approach is Bayesian, it is expected to extract information from every single photon, as evidenced by the fact that the method still works at the fundamental limit. In this respect, it currently outperforms all competing methods.

Concentration-dependent protein aggregation is a common pitfall of scattering experiments (i.e. SAXS/SANS; not necessarily of successful crystallizations though), and a protein/protein construct that suffers from it will be doomed in terms of structure determination by XFEL scattering as well. Theoretical predictions can sometimes be too optimistic in terms of the experimental possibilities: phosphorus SAD has run into a wall but sulfur phasing has worked out beyond most expectations in crystallographic phasing; neutron diffraction has not lived up to its promises even as higher flux SNS beam lines have come online and crystal size remains a limitation; etc.

Interestingly, as also noted in our reply to Reviewer #1 and #2 concerning hit-rates, we assume aggregation to be a far lesser problem for our method than, e.g., it is for SAXS/SANS. We have not been aware of this strength before, and thank the Reviewers for directing our reasoning into this exciting new direction. **We have now added a paragraph explaining this issue to the discussion (p. 16), as well as new section to the Supplement explaining and supporting this claim.**

Nevertheless, despite these concerns, the present manuscript reporting the potential benefits and future possibilities of XFEL scattering for structural biology are pretty exciting and worth publishing.

Many thanks for the constructive comments and the final recommendation.

Minor corrections:

1) Page 11, 4 lines from bottom: which corresponds.

2) Page 13, 5 lines from end of text: *relevant (relevant)*.

Corrected.

Reviewers' comments:

Reviewer #1 (Remarks to the Author):

I think the authors did a good job at addressing all the referees issues. The only small annoying issue is the calculation of the coherent scattering cross section. I would suggest the authors to contact a local expert, such as Tim Salditt to clear this point, as it's a stain in an otherwise nice paper. Regardless I suggest accepting it for publication.

Reviewer #2 (Remarks to the Author):

I commend the distinguished authors for the revisions to the manuscript.

Leaving aside detailed questions for the moment, the difficulty is this: A case can be made that the primary claims of the paper have been exceeded by prior published work.

- The present paper claims to have reached the limit of 3 photons per frame. This limit pertains only to three-point correlation methods, and the demonstration is based on simulated data. Already six years ago, Philipp et al. (1, 2) demonstrated successful structure recovery from experimental diffraction patterns containing an average of 2.5 photons per frame.
- The present paper claims atomic resolution with simulated data. This has been previously demonstrated, for example by Fung, et al. (3) for a small molecule, and by Hosseinizadeh et al. (4) for a virus.

These concerns, raised in my previous review, unfortunately still stand.

Perhaps I may suggest a way forward. In the revised manuscript, I was encouraged to see results obtained with the experimental data published by Reddy et al. (5). The "success" of the outcome, however, has not been quantified, and the reconstruction shown in Supplementary Fig. 6 does not seem overly impressive. I recognize, however, that the photon count in the Reddy dataset is far higher than the optimal range for the new algorithm. This problem can be easily circumvented by randomly down-sampling the number of photons in each experimental frame of Reddy et al. to an average of three. A quantitative assessment of the performance of the new algorithm in its optimal signal range would then demonstrate its true potential.

Specific Comments

1. The issues of computational scaling and expense remain to be adequately treated. The computational time and memory requirements for an atomic-resolution reconstruction of a reasonable size (say, 50 - 100kDa) object, and the way these computational requirement scale with key parameters (ratio of object diameter to resolution, number of diffraction patterns, photon count per frame and in total, signal to noise ratio) need to be clearly laid out. A quantitative discussion of these issues (including simulated annealing) is essential to assess the potential of the approach.
2. The average of correlations does not in general equal the correlation of the averages. How does this impact the present method, particularly if the particles are not identical?
3. A clear discussion of how independent samples were drawn for analysis by Fourier Shell Correlation (FSC), and a plot of the FSC would be helpful.
4. Reconstruction(s) of the PR772 by three-point correlations should be quantitatively compared with other published reconstructions from the same data (6, 7).
5. The signal-to-noise ratio should be clearly defined and specified for each of the examples.

6. A pseudocode of the algorithm should be included in the supplementary materials.
7. The manuscript includes, in my view, inaccurate and unsubstantiated statements. A few examples:
 - a. "Standard methods cannot cope with the high statistical noise in this extreme Poisson regime" (p.3). (Inaccurate in the light of (1).)
 - b. "[These algorithms] are prone to instability in the presence of noise" (p. 4). (No citation supporting the assertion.)
 - c. "[...] we have successfully determined the structure of the coliphage PR772 [...]" (p. 15). (What constitutes "success"? What are the quantitative criteria? What resolution was achieved? Etc.)

I am certain the distinguished authors will wish to identify and amend all such statements.

References:

1. Philipp HT, Ayyer K, Tate MW, Elser V, Gruner SM (2012) Solving structure with sparse, randomly-oriented x-ray data. *Optics Express* 20(12):13129-13137.
2. Philipp HT, Ayyer K, Tate MW, Elser V, Gruner SM (2013) Recovering structure from many low-information 2-D images of randomly-oriented samples. *Journal of Physics: Conference Series* 425(19):192016.
3. Fung R, Shneerson V, Saldin DK, Ourmazd A (2009) Structure from fleeting illumination of faint spinning objects in flight. *Nature Physics* 5(1):64-67.
4. Hosseinizadeh A, et al. (2014) High-resolution structure of viruses from random diffraction snapshots. *Phil Trans R Soc B* 369(1647):20130326.
5. Reddy HKN, et al. (2017) Coherent soft X-ray diffraction imaging of coliphage PR772 at the Linac coherent light source. *Sci Data* 4:170079.
6. Kurta RP, et al. (2017) Correlations in Scattered X-Ray Laser Pulses Reveal Nanoscale Structural Features of Viruses. *Physical Review Letters* 119(15).
7. Hosseinizadeh A, et al. (2017) Conformational landscape of a virus by single-particle X-ray scattering. *Nature methods* 14(9):877-881.

Reviewer #1

I think the authors did a good job at addressing all the referees issues.

Many thanks!

The only small annoying issue is the calculation of the coherent scattering cross section. I would suggest the authors to contact a local expert, such as Tim Salditt to clear this point, as it's a stain in an otherwise nice paper.

Regardless I suggest accepting it for publication.

We wholeheartedly agree, this issue needed to be settled, as the uncertainty about the expected number of detected photons per image is uncomfortable and should not be as large as written in the first revision – where we did not discuss this issue further, because none of our conclusions really rests on it. In fact, already the Referee's original comment made us dig deeper, and we fully agree that the coherence terms increase our original simple estimate. Because this will allow to use somewhat lower beam intensities or larger beam diameters as described below, this amendment will further broaden the applicability of our method.

As suggested by Referee #1, we now use Condor for our estimate and have calculated photon count estimates for different reported beam diameters ranging from:

d = 100 nm (Tegze2012): 1356

d = 500 nm (Daurer2017): 54

d = 1000 nm (Condor with LCLS data from previous dwarf virus): 14

For the estimates above, we have now used a still conservative $1e12$ photons/beam (our initial flux of $5e11$ photons/beam was below all reported expected fluxes in the literature for next-generation FELs). All results were confirmed by estimates obtained from the SimEX framework using realistic beam profiles provided by Yoon from XFEL Hamburg.

We are very pleased with this result (also discussed with and cross checked by Tim Salditt, as suggested), as it implies that lower beam fluxes (i.e., less technically demanding focal sizes) and/or much smaller number of images are required than (conservatively) stated in our original manuscript, bringing our approach even closer to experimental reach.

We have now updated the examples in our revised manuscript accordingly.

Reviewer #2

I commend the distinguished authors for the revisions to the manuscript.

Thank you! Indeed, we would like to thank the Referee for the many helpful comments, which helped us to further improve and strengthen the manuscript. The Editor has, thankfully, allowed to reply in a second (last) round to the Referee's comments, so we hope the Referee will agree that our work represents a novel and substantial step forward, albeit it may not provide an answer to every single question that may come up during experimentation in the years to come.

Leaving aside detailed questions for the moment, the difficulty is this: A case can be made that the primary claims of the paper have been exceeded by prior published work.

- *The present paper claims to have reached the limit of 3 photons per frame. This limit pertains only to three-point correlation methods, and the demonstration is based on simulated data. Already six years ago, Philipp et al. (1, 2) demonstrated successful structure recovery from experimental diffraction patterns containing an average of 2.5 photons per frame.*

- *The present paper claims atomic resolution with simulated data. This has been previously demonstrated, for example by Fung, et al. (3) for a small molecule, and by Hosseinizadeh et al. (4) for a virus.*

These concerns, raised in my previous review, unfortunately still stand.

We thank the Referee for now specifying two of the 'cases at extremely low photon counts' he/she had in mind in the first report. We also very much appreciate the Referee's suggestion of how to resolve the issue, also concerning validation of our method with real experimental data of Coliphage PR772. As detailed further below, we have followed this suggestion and implemented the proposed test. As is now described in the revised manuscript and further below, we have also demonstrated and assessed our structure determination for this case.

We respectfully disagree with the Referee's claim that '*the primary claims of the paper have been exceeded by prior published work*'.

Specifically, using the well-known expectation maximization algorithm, Philipp et al. (1, 2) only address the problem of a *two-dimensional* (i.e., only one unknown rotation angle) *macroscopic* object rather than the much more demanding case of a three-dimensional *molecule* with *three* unknown orientation angles. In Ref. (2) they announce in their last paragraph 'to test the ability of similar recovery of a 3D structure with one axis of rotation', but still only show low-resolution projections of a macroscopic object from 99 photons per image in Ref. [Ayyer, K., Philipp, H. T., Tate, M. W., Elser, V., & Gruner, S. M. (2014). *Real-Space x-ray tomographic reconstruction of randomly oriented objects with sparse data frames. Optics Express, 22(3), 2403*], with no quantitative quality assessment provided. Contrary to the claim of the Referee, up to now, no 3D reconstruction from images with arbitrary 3D orientations for the low (average) photon number of 2.5 have been reported. In light of the subsequent literature (reviewed in our introduction), which consistently reports successful *de novo* 3D structure determination only for well over 100 photons per frame, we conclude that, as of now, the expectation maximization algorithm does not exceed our

primary claim. To avoid potential misunderstanding, we now mention the achievement by Philipp et al. (1, 2) in our revised manuscript.

Also Fung, et al. (3) and Hosseinizadeh et al. (4) do not support the Referee's claim: Both works, as well as all others we're aware of, require well above 100 photons per frame, as we have already explained and cited in our original manuscript.

Perhaps I may suggest a way forward. In the revised manuscript, I was encouraged to see results obtained with the experimental data published by Reddy et al. (5). The "success" of the outcome, however, has not been quantified, and the reconstruction shown in Supplementary Fig. 6 does not seem overly impressive. I recognize, however, that the photon count in the Reddy dataset is far higher than the optimal range for the new algorithm. This problem can be easily circumvented by randomly down-sampling the number of photons in each experimental frame of Reddy et al. to an average of three. A quantitative assessment of the performance of the new algorithm in its optimal signal range would then demonstrate its true potential.

We very much appreciate this suggestion, and have now implemented it one-to-one in our manuscript, demonstrating structure determination also for this artificially down-sampled low photon count case. As also suggested by the Referee, we have now quantitatively assessed the resolution of the obtained density by including appropriate Fourier shell correlations (See Supp.X). As can also be seen from our revised Supp. Fig.7 & Fig.8, the resolution of the reconstructed density is only slightly lower (11.3 nm vs. 9 nm) than the one published in Ref.7, Fig.1. Importantly, it nicely shows the icosahedral symmetry of the virus. Addressing the previous remark of the Referee that "[the reconstruction] is not overly impressive", we have contacted the author of Ref.7, A. Ourmazd, and learned that in their reconstruction, icosahedral symmetry was implicitly enforced by using icosahedral Wigner D functions [A. Ourmazd, personal communication], whereas no such symmetry was enforced in our reconstruction. We attribute the slightly lower resolution we achieved mainly to that very fact. Further, we had only access to a (publicly available) image set of 14,722 images [<http://cxidb.org/id-58.html>], whereas a set of 37,550 slightly differently filtered images has been used [Ref.5] for Fig.1 in Ref.7. In light of these considerations, we think the Referee will agree that the quality of our reconstructed density is indeed sufficiently high.

As can also be seen from our new data, the suggested down-sampling did not decrease the quality of our reconstruction. In fact, we are not surprised by this outcome: Because our method uses triples of registered photons as input (rather than frames), it does not even 'know' from how many frames (and from which frame) the photon-triples were extracted and, hence, the suggested down-sampling is essentially not even seen by our method. 'Essentially', because there are two amendments to be considered. First, triples collected from the same frame may show (higher order) correlations (because they arise from the same orientation of the sample), which are absent for triples collected from separate frames. These correlations may contribute to the statistical noise of the three-photon histograms and, hence slightly reduce the resolution. And indeed, as now shown in the revised manuscript, the suggested random down-sampling even slightly improved the quality of our reconstruction. The second amendment is more technical in nature. As we had described in our first revision, we have used as a computational shortcut direct calculation of the three-photon histograms from the intensities reported in Reddy et al. (5), rather than from accumulation of individual photon triplets. Because this short-cut may have reduced the statistical noise in the three-photon histograms, we have now repeated the calculation using randomly down-sampled frames, precisely as the Referee suggested.

We thank the Referee for this suggestion, which removed these two amendments, yielded even better results, and thereby further strengthened our manuscript.

Specific Comments

1. The issues of computational scaling and expense remain to be adequately treated. The computational time and memory requirements for an atomic-resolution reconstruction of a reasonable size (say, 50 - 100kDa) object, and the way these computational requirements scale with key parameters (ratio of object diameter to resolution, number of diffraction patterns, photon count per frame and in total, signal to noise ratio) need to be clearly laid out. A quantitative discussion of these issues (including simulated annealing) is essential to assess the potential of the approach.

We agree with the Referee that – computational cost being significant – the main scaling properties of our method will also be of interest to the broader readership of Nature Communications and have therefore now been included.

As is now described in the revised manuscript, the computational cost

(1) scales as approx. n^7 with ratio of object diameter to resolution, which for given desired resolution translates into $M^{2.33}$ with M the molecular weight,

(2) is independent of the number of images, as the assembly of the histograms is not computationally limiting.

(3) Signal to noise ratio does not directly affect computational cost, but of course reduce resolution, as exemplified in Fig.3.c.

(4) Memory requirements are low and not an issue.

(5) Also the number of simulated annealing steps (or similar, for improved optimization methods) will likely increase with molecular weight, although no scaling laws can be derived. Empirically, we found an approximately $m \log m$ scaling with the number $m = 2L+1$, as stated in the supplemental Information V.B₄, which in turn scales linearly with molecular weight.

As is now summarized in the revised manuscript, we expect computational effort to scale slightly faster than the molecular weight cubed – underscoring our statement in the Conclusion of the original manuscript that computational resources may become a limiting factor for much larger molecules at high resolution. Even without considering further algorithmic and numerical improvements, however, a tiny fraction of the total budget of the experiments would allow to set up sufficiently many GPUs to grossly alleviate this issue.

Let us also emphasize that, as of now, our current implementation is not highly optimized for computational efficiency and scaling, which will require significant further implementation efforts. This is particularly true for the used optimization method, which leaves much room for computational improvements. Hence, any benchmarks we provided would likely have to be revised (towards higher computational efficiency) as soon as more efficient software implementations become available. Given the highly competitive situation in the field, however, as well as the general readership and scope of Nature Communications, we would like to avoid the considerable delay of publication such implementation and benchmarking work would imply, and would prefer to follow up with technical details on algorithmic efficiency improvements and benchmarking of computational costs and in a subsequent publication in a more specialized Journal.

2. The average of correlations does not in general equal the correlation of the averages. How does this impact the present method, particularly if the particles are not identical?

Whereas it is true that in general the average of correlations does not in general equal the correlation of the averages, in the case of a mixture of several different particles, the three-photon correlation nevertheless is the linear superposition of the three-photon correlations of the individual species. This is because each photon triplet originates from one particle, such that in this case no cross terms arise. Important prerequisite is that no more than one particle is hit at a time, underscoring that single particle experiments are essential here. (Note that the situation is different if several particles are in the beam, in which case cross terms do arise, as is now discussed in the Supp.IX.). Therefore, as in x-ray crystallography, structural heterogeneity will generally either reduce signal and resolution, or – given sufficient data and resolution – can be treated via structural sub-populations, i.e., a superposition of several different structures or conformations. The latter should be possible in a conceptually straightforward way, albeit at probably significantly increased computational and experimental cost. Given the many uncertainties in light of absent experimental data, we would very much prefer not to overload our manuscript with a detailed treatment. Instead, we have now refined our discussion appropriately.

3. A clear discussion of how independent samples were drawn for analysis by Fourier Shell Correlation (FSC), and a plot of the FSC would be helpful.

For Crambin, such discussion and respective results are already shown in the previous version. We have now also included this information for the Coliphage as stated above.

4. Reconstruction(s) of the PR772 by three-point correlations should be quantitatively compared with other published reconstructions from the same data (6, 7).

We agree and have now included a comparison via correlation coefficient with the data from Ref (6,7) and well as appropriate FSC plots within the Supplementary Information.

5. The signal-to-noise ratio should be clearly defined and specified for each of the examples.

We have now made sure the signal-to-noise ratio is clearly defined in each case, also emphasizing the fundamental difference between shot noise and additional sources of noise.

6. A pseudocode of the algorithm should be included in the supplementary materials.

We have now added a flow chart to this effect within the Supplementary Information Fig.1, which in our view provides a clearer overview for the interested reader. We feel that a full pseudocode would not be more helpful, and also not much different or shorter than our well-documented ThreePhoton.jl julia library (see file determination.jl), which we have made publicly available.

7. The manuscript includes, in my view, inaccurate and unsubstantiated statements. A few examples:

a. *“Standard methods cannot cope with the high statistical noise in this extreme Poisson regime” (p.3). (Inaccurate in the light of (1).)*

We refer to our discussion above and, to avoid misunderstanding on the readers' side, have now modified this statement to “The high statistical noise in this extreme Poisson regime poses considerable methodological challenges and, hence, XFEL structure determination attempts resorted to nano-crystals almost exclusively focus on nano-crystals.”

b. *“[These algorithms] are prone to instability in the presence of noise” (p. 4). (No citation supporting the assertion.)*

We have removed this sentence.

c. *“[...] we have successfully determined the structure of the coliphage PR772 [...]” (p. 15). (What constitutes “success”? What are the quantitative criteria? What resolution was achieved? Etc.)*

As requested and discussed above, we have now included and discuss quantitative comparisons of FSCs within our revised manuscript.

REVIEWERS' COMMENTS:

Reviewer #2 (Remarks to the Author):

I strongly commend the distinguished authors for the excellent revisions. While experts can discuss various points over a suitable beverage, I believe the paper is now of sufficiently high quality and interest to warrant publication in Nature Communications.

I do not wish to delay things further, but think the following points need to be incorporated into the final manuscript.

1. The experimental snapshots used to reconstruct the PR772 coliphage by three-point correlations were specially selected by a sophisticated approach. It is important to point this out to prevent the impression that the same results would have been obtained from a random selection of experimental snapshots.
2. A key determinant of spatial resolution is the number of snapshots, with the application of icosahedral symmetry playing only a minor role. It is a little dangerous to ascribe the lower resolution simply to imposition of symmetry (see p. 15 of manuscript).

Reply to Reviewer #2:

I strongly commend the distinguished authors for the excellent revisions. While experts can discuss various points over a suitable beverage, I believe the paper is now of sufficiently high quality and interest to warrant publication in Nature Communications.

I do not wish to delay things further, but think the following points need to be incorporated into the final manuscript.

Many thanks for the helpful comments throughout the review process.

1. The experimental snapshots used to reconstruct the PR772 coliphage by three-point correlations were specially selected by a sophisticated approach. It is important to point this out to prevent the impression that the same results would have been obtained from a random selection of experimental snapshots.

This is correct. We now mention the diffusion map embedding at the end of the Results section.

2. A key determinant of spatial resolution is the number of snapshots, with the application of icosahedral symmetry playing only a minor role. It is a little dangerous to ascribe the lower resolution simply to imposition of symmetry (see p. 15 of manuscript).

We agree and added that the slightly reduced resolution may also be attributed to the fact that we used much fewer photons than the Reference, implying additional Poisson noise.